# Increasing Machine-Related Safety on Farms: Development of an Intervention Using the Behaviour Change Wheel Approach

**DOI:** 10.3390/ijerph20075394

**Published:** 2023-04-04

**Authors:** Aswathi Surendran, Jennifer McSharry, Oonagh Meade, Francis Bligh, John McNamara, David Meredith, Denis O’Hora

**Affiliations:** 1School of Psychology, University of Galway, H91 TK33 Galway, Ireland; s.aswathi@nuigalway.ie (A.S.);; 2Irish Agriculture and Food Development Authority, R93 XE12 Carlow, Ireland

**Keywords:** behaviour change intervention, farm safety intervention, tractors, peer-to-peer mentoring, COM-B, BCT, occupational safety and health, blind spots

## Abstract

Farming is essential work, but it suffers from very high injury and fatality rates. Machinery, including tractors, are a leading cause of serious injuries and fatalities to farmers and farm workers in many countries. Herein, we document the systematic development of an evidence-based, theory-informed behaviour change intervention to increase machine-related safety on farms. Intervention development progressed through four phases. Phase 1 defined the problem in behavioural terms based a review of the literature, Phase 2 identified candidate intervention targets through a series of focus groups guided by the Capability–Opportunity–Motivation–Behaviour (COM-B) model and Phase 3 employed expert and stakeholder consultation guided by the Behaviour Change Wheel (BCW) to consider potential target behaviours and intervention components and finalise the intervention content. Phase 4 finalised the evaluation strategies with a team of agricultural advisors who supported the rollout and identified outcome measures for the first trial. The target intervention was the identification of blind spots of farm tractors, and three priority target behaviours (farm safety practices) were identified. Following Phase 3, the intervention comprised four components that are delivered in a group-based, face-to-face session with farmers. In Phase 4, the acceptability, feasibility, and fidelity of these components were identified as the outcome measures for the first trial of the intervention. The four-phase systematic method detailed here constitutes an initial template for developing theory-based, stakeholder-driven, behaviour-change-based interventions targeting farmers and reporting such developments.

## 1. Introduction

Despite the global effort to improve farm safety, injury and fatality rates remain high in the agricultural sector [1,2,3,4]. Farmers comprise only six percent of the Irish working population; however, the agriculture sector reports approximately half of the occupation-related fatalities [5,6]. The Irish Agriculture and Food Development Authority (Teagasc) National Farm Survey reported a 31% increase in farm incidents in the last decade in Ireland [7]. Furthermore, Mohammadrezaei et al. [8] observed that farm injury is more likely to lead to severe injuries and fatalities than other work-related injuries. More than half of the reported injuries occur from farm machines, vehicles and livestock. Tractors are linked to 55% of all vehicle work-related fatalities and 25% of reported injuries [9]. Given the global concern about persisting high fatality rates on farms, improving the safety of farms is a key health and safety policy issue. Policy-makers and researchers have therefore emphasised the importance of developing effective and affordable interventions to improve the safety behaviours of farmers [10,11,12].

The Medical Research Council (MRC) guidance for developing and evaluating complex interventions advocates for a systematic approach involving the best available evidence and appropriate theories [13]. Despite the growing evidence on the significant role of behavioural science in developing comprehensive injury prevention strategies, farm safety research has lagged behind other industries in the use and reporting of behavioural strategies [6,12]. Historically, farm research has relied on introducing technological and regulatory interventions and educational interventions focusing on informing farmers of the risks on farms [14,15,16]. However, the normalisation of the danger and persisting risky farm practices indicate a risk awareness to risk prevention behaviour gap [17,18,19]. The gap suggests that focusing on improving awareness alone cannot mitigate the potential risks on farms. Studies suggest that farmers’ behaviour is a product of the interaction between cognitive factors, such as perceived efficacy and beliefs, and environmental and technological factors, such as the size of the farm and the type of machinery available [6,20]. Hence, recent reviews of the farm safety interventions call for research focusing on understanding the factors that influence farmers’ safety behaviours and developing interventions targeting these factors [12,14,15].

A recent systematic review by the authors indicated that there are growing numbers of studies utilising behaviour change theories such as the Theory of Planned Behaviour [21] and the Health Belief Model [22] to understand the factors influencing farmers’ behaviours [23,24]. However, safety literature indicates that intervention studies often fail to report the use and role of underlying behavioural theories in intervention development. They often provide little information on the intervention developments, its components and delivery, and even less on how the individual components influenced the target behaviour(s) [12,14,25]. The lack of empirical evidence on the active ingredients of these interventions and lack of focus on specific target behaviours (farm practices) make it difficult to draw a conclusion on what part of the intervention worked and how it worked [26,27,28]. This lack of clarity also reduces the potential for the interventions to be replicated or adopted.

Recent advancements in behavioural science have resulted in tools and techniques to develop interventions in a more systematic, evidence-based, theoretically informed way [29]. The objective of the current study is to use the systematic Behaviour Change Wheel (BCW) intervention development approach to use existing evidence, and farmer and other stakeholder perspectives to develop an evidence-based, theoretically informed intervention to increase machine-related safety on farms.

### Intervention Development Process

This study is part of a larger, multi-phase BeSafe project aimed at addressing the limitations of previous safety interventions targeting machine safety on farms. The study aimed to address gaps in the literature by developing an intervention using a systematic approach informed by the most relevant evidence, appropriate theories and stakeholder engagement.

Previous studies exploring farmers’ safety behaviour employed behaviour models such as the Theory of Planned Behaviour (TPB), Capability, Opportunity, Motivation–Behaviour (COM-B) model and Health Belief Models to explain how farmers’ intentions, beliefs and attitudes are developed and evolve through the interactions between internal and external factors [23,30,31,32]. One framework that has gained popularity in health research for intervention development is the Behaviour Change Wheel (BCW) [33]. The BCW framework was developed by synthesising 19 existing behaviour change frameworks and provides a systematic, comprehensive approach for diagnosing who (target population) needs to perform what (target behaviour) and which behaviour determinants (barriers and facilitators) need to be targeted by what type of intervention content [34]. Evidence from the health research indicates that BCW can provide guidance on mapping farmer-centric determinants into the existing behaviour change constructs and subsequently providing recommendations for the behaviour change components relevant to specific constructs and operationalising the intervention contents [27,34,35,36].

As illustrated in Figure A1, at the hub of the wheel is the COM-B model, which describes behaviour as a function of capability (physical/psychological), opportunity (social/physical), and motivation (reflective/automatic). According to the BCW, the intervention must target one or more of these components to promote the desired behaviour. The next layer of the BCW outlines nine intervention functions that describe a broad category of interventions (education, persuasion, incentivisation, coercion, training, enablement, modelling, environmental restructuring and restrictions) [34]. These are then mapped to behaviour change techniques (BCTs), the observable, irreducible, replicable “active ingredients” of interventions. The BCT Taxonomy v1 [37] is a structured list of 93 BCTs with definitions. Once the BCTs are identified, the next step is operationalising the BCTs and identifying potential ways to put the selected BCTs into practice [33,37]. BCTs can be combined to form intervention components. The final intervention package may comprise different intervention components and act upon one or more mechanisms of behaviour mediators [29,34,35,38].

This paper outlines the systematic process used to develop the BeSafe intervention and constitutes a template for developing similar interventions and reporting development decisions. This paper describes (a) the intervention development process (b) the relevant output of phase 1–3 and how it was methodically mapped to the BCW framework, and (3) the content of the resulting intervention. It will not discuss in detail the evaluation strategies and outcomes of the evaluations as they are outside the scope of the current paper and will be published separately in detail.

Reporting of the intervention is in accordance with the TIDieR (Template for intervention description and replication) guidelines [39] and is available in the Appendix A (Table A5). The authors also considered the best practice guidelines [40] on reporting intervention development. Therefore, the rationale for the inclusion of behaviour change theory, the inclusion of existing evidence, the contribution of stakeholders, the modification of intervention components, etc., are included.

## 2. Method

The BeSafe project is a research programme funded by the Department of Agriculture, Ireland, and supported by the Health and Safety Authority (HSA) and Teagasc (the Irish state agency providing research, advisory and education in agriculture) to develop safety interventions to bring long-term changes to machine-related safety on farms. The current study commenced in October 2019. The research team consisted of a doctoral student (AS), two behavioural researchers (DOH. and JMS) and a Teagasc advisory team (FB, JMN and DM). The development of the BeSafe intervention involved four key phases, as illustrated in Table 1.

Intervention development was an iterative process guided by the intervention development guidelines provided by the BCW framework [34], including the findings from the farm safety literature, stakeholder recommendations and expert opinions. As illustrated in Figure A2, the intervention development process has been broadly categorised into three stages over eight steps [33]. As illustrated in Table 1, the current study had four phases, including the evaluation strategy development. Phase 1 described the problem in behavioural terms through a review of the evidence, Phase 2 identified what needs to be changed through a series of focus groups, Phase 3 explored the potential target behaviours and intervention components through expert and stakeholder consultation and finalised the intervention content, and Phase 4 finalised the evaluation strategies with the Teagasc advisory team.

Ethical approval was obtained from the Research Ethics committee (REC) of the National University of Ireland, Galway (NUIG), before the commencement of focus groups (#2020.10.022) and co-design workshops (#2021.01.013). Written/verbal consent was obtained from all the participants involved in the study.

The current paper was built on the evidence generated from the first three phases and described how the findings from these phases were methodically mapped onto the BCW framework and BCTTv1 to create lists of potential target behaviours and intervention components to address them. In this paper, the authors will explain how a behavioural change-based intervention to adopt preventive safety behaviours to address tractor-related blind spots at farms and improve farm safety was developed.

### 2.1. Phase 1: Describe the Problem in Behavioural Terms

The first phase involved identifying and analysing the existing relevant evidence base, and examining the injury and fatality reports [41] to understand farm safety in the context of Irish farms. A systematic review of the interventions targeting machine-related injuries safety on farms had four objectives:
What interventions have been employed to reduce machine-related incidents, injuries and fatalities among farmers?How effective are interventions designed to improve machine-related safety on farms?What BCTs and intervention functions comprise these interventions?What are the gaps in the current interventions?


A systematic review protocol was developed per the Preferred Reporting Items for Systematic Review and Meta-Analysis Protocols (PRISMA-P) guidelines and registered (Registration number: CRD42020173834) on PROSPERO, the International Prospective Register of Systematic Reviews [42].

In the review, as the first step, the general intervention categories were identified, such as safety education, financial assistance, regulations, etc. On further analysis, the behaviour change components targeting machine safety present in these interventions were identified and coded. Following the mapping of the components, the BCT used to implement these intervention functions was coded using the BCT taxonomy V1 [43]. A narrative synthesis of the evidence was conducted due to the heterogeneity of the included studies. 

Reports of farm injuries and practices from HSA were examined to identify the potential broad categories of target population and behaviours for the intervention [41]. Reviewing the ground report along with the safety literature assisted the authors in identifying the existing conditions, tendencies and gaps within the safety intervention literature. A detailed description of the findings from the systematic review will be published separately.

### 2.2. Phase 2: Identify What Needs to Be Changed and Which Barriers and Enablers Need to Be Addressed?

In the next phase, between January and February 2021, we conducted a qualitative study involving four semi-structured focus groups with Irish farmers above 60 years of age. Purposive sampling [44] was adopted to ensure that farmers from all four farm types (dairy, beef, sheep and tillage) were represented. A topic guide informed by the COM-B model [34] and review findings was used flexibly to identify the target population and guide the focus groups to explore barriers to and facilitators to adopting safe practices linked to tractors and quad bikes. The study explored capability-related barriers, such as a lack of knowledge and ability to manage the demands of farm work; opportunity-related barriers, such as access to resources and market conditions; and motivation-related barriers, such as beliefs about the benefits of taking risks and perceived self-efficacy. The objectives of the focus group discussions were to:
5.Explore participants’ experiences of tractor and quad bike safety.6.Identify the potential barriers and facilitators to safety behaviours.7.Identify potential active components to address these behaviours.


A series of four focus group discussions conducted online via Skype [45,46] by the primary author had a total of 19 participants aged above 60, representing the four major farm systems of Ireland. Data were analysed inductively using thematic analysis [47] with MAXQDA 2020 software [48]. Inductively generated themes were then mapped to the COM-B domains [34]. A detailed description of the findings from the study will be published separately. 

### 2.3. Phase 3: Identification of Potential Target Behaviours and Intervention Components

Before the commencement of this phase, the research team met to review Phase 1 and 2 findings and summarise the relevant findings. Subsequently, the potential target behaviours identified by the farmers were analysed and grouped into nine categories. The categorisation was guided by the findings from the review, survey reports [41] and safe work practice guidelines [5]. Categories and their breakdown are detailed in Table A1. 

The current phase involves two steps: 1. Two co-design workshops with safety experts and stakeholders and 2. Feasibility screening with Teagasc advisory team.

#### 2.3.1. Phase 3.1 Co-Design Workshop

The current phase involved two stages involving one co-design workshop each, facilitated by AS, DOH and JMS. The session was conducted and recorded via the video conferencing tool, Zoom [49], with five to six participants per session. The objectives of these workshops were:
8.Identify potential target behaviours for the intervention.9.Identify barriers and enablers that are likely to influence these target behaviours. 10.Identify the potential intervention components and delivery methods.


As a first step, international farm safety experts with expertise in farm safety and farmer behaviours and stakeholders such as safety inspectors, farm representatives, tractor dealers and other farm organisation representatives were invited to participate in the workshops. They were assigned to one of the two sessions based on their expertise and availability. 

Identification of target behaviours

The research team created a web-based rank order survey using the Gorilla survey builder [50]. The survey included the nine categories of potential target behaviours as the survey items. The influence of these behaviours on fatal farm incidents among farmers aged over 60 from 2004 to 2018 and the examples of fatal incidents involving these behaviours were provided in the survey for reference (Table A1). A week before the co-design workshop, the research team shared the summary of findings from the previous phases with the participants of both workshops via email. They were also invited to identify the top categories of target behaviours to be considered for the workshop. Hence, a link to the survey was shared with them, and they were asked to place the items mentioned in rank order of relative importance. To determine the importance of each category, they were asked to consider the expected impact on the safety of farmers over 60 years old using farm tractors and machinery and whether these behaviours can be addressed effectively through behavioural intervention. The online workshop session began by sharing the findings from the web survey. The participants were informed that the top two categories would be considered for the current discussion, and they were encouraged to explore the specific safety behaviours under the top two categories that can be considered for a behaviour-based intervention. As the discussion progressed, the focus shifted to identifying the relevant barriers and facilitators to be considered. They were encouraged to prioritise the barriers and facilitators that met the following criteria:
11.Relevance to older Irish farmers.12.Effectively addressable through behavioural interventions.13.Feasibility to address them within the project constraints such as funding and available time. 14.Significant influence on risky farm practices.15.Influence more than one risky farm practices.


For example, extensive paperwork as a barrier was given low priority since that is more relevant to policy-based or organisational-level intervention than behaviour-based intervention. Likewise, whenever a participant suggested a barrier or facilitator, facilitators encouraged them to discuss its impact on the older farmers, how it impacts them and whether it influences more than one safety behaviour. For example, farmers indicated that having a succession plan not only reduces the workload but also motivates investment in safety. However, regardless of meeting the criteria, all the barriers and facilitators identified by the workshop will be made available separately for future reference.

Identification of potential intervention strategies and delivery methods.

After the first workshop, the research team met to analyse and summarise the relevant findings. The second workshop had a different set of participants but with similar expertise and knowledge. The key evidence from previous phases, along with the recommendations formulated based on the BCW framework, was presented to the workshop participants. Participants were then asked to narrow down the potential barriers and facilitators to be addressed by the intervention and identify the potential intervention strategies and delivery methods to address them. 

#### 2.3.2. Phase 3.2: Feasibility Screening with Teagasc Advisory Team

The objectives of the screening were to finalise the selection of target behaviours, behaviour change techniques and modes of delivery.

Once the key recommendations from the previous phases were consolidated (potential list of target behaviours, intervention strategies and delivery methods) and mapped to the BCW framework, the research team added their ideas to the list of potential modes of delivery for each BCT. 

From the selected target behaviours, specific target behaviours were selected and finalised for the intervention based on the following criteria:
16.Availability of evidence on the influence of the target behaviour on fatal incidents.17.Potentially modifiable at the farmer level.18.Farmers’ ability to carry out regardless of their age.19.Part of the recommended safer practices guidelines for operating the tractor safely.20.Relevant to every tractor-operating farmer regardless of age and farm type.


Once the target behaviours were finalised (see Table A2), the focus shifted to identifying the active ingredients and their delivery strategies from the consolidated list. The Acceptability, Practicability, Effectiveness, Affordability, Side-effects, and Equity (APEASE) [33,51] criteria were applied to remove the least preferred strategies. The authors prioritised a subset of the criteria; acceptability, practicability and affordability over others (effectiveness, spill over effects/safety, equity) that are more relevant for the full-scale trial [36]. The team tested the criteria as follows:

Acceptability: How likely will the farmers engage with the activity and were these intervention techniques familiar to them?

Practicability: How likely is it to be completed in the allocated time, how much training is required, is it safe to perform, how many facilitators are required and can this be transitioned well for the large-scale rollout in the future?

Affordability: How likely the interventions can be implemented within the allocated budget.

Effectiveness: What are the expected outcomes of the trial? Do these intervention components effectively educate the participants on completing the target behaviours at home?

Side effects/Safety: What are the other farm practices likely to be influenced by these interventions? Are there any negative effects expected to arise from the intervention?

Equity: How far the intervention or part of the intervention likely to affect equity of access?

After reaching a consensus on the finalised intervention components and implementation strategy within the BeSafe study team, the potential evaluation strategies were discussed. These discussions also determined the resources and funds available for the implementation and identified the ethical and bureaucratic approvals required for it. 

### 2.4. Phase 4: How Can Behaviour Change Be Measured and Understood?

While the systematic and transparent reporting of methodology serves as a guide for developing interventions, a comprehensive evaluation is necessary to examine the feasibility and efficacy of the proposed behaviour change components. The systematic and transparent results presented in this paper will aid in conducting a thorough evaluation of the key components and their effectiveness. It was determined that the focus of the first trial would be on assessing the feasibility, fidelity and acceptability of the intervention components and their delivery. 

At the final stage, the outcome measures to evaluate the feasibility and determine the behaviour change, which included the completion of target behaviours, were determined. The finalisation of the evaluation was guided by a pre-determined feasibility checklist, fidelity framework and theoretical framework of acceptability, respectively, to ensure a systematic evaluation of the intervention. The following outcome measures were proposed:
An intervention checklist, Direct observation, Audio recording of the study, Reported experience of the facilitators, Exit survey, SMS survey, Personal interviews. 


Evaluation strategies were identified for their potential to measure the feasibility, fidelity and acceptability of the active ingredients of the intervention, their delivery methods as well as the intervention as a whole. The tools for measurement were selected based on reliability, validity, availability and relevance. The potential list of recommended strategies was also judged against the subset of APEASE criteria considered previously. Other considerations were the suitability of digital tools such as online surveys for older participants and high attrition rates reported in farm interventions. 

Based on the findings from the study, a large-scale effectiveness trial will be recommended for the future stages of the intervention evaluation. 

## 3. Results

### 3.1. Phase 1: Describe the Problem in Behavioural Terms 

The systematic review reiterated the findings of the previous reviews on the lack of theoretically informed behaviour change interventions, limiting the sustainability and efficacy of interventions [4,12]. Although reviewed studies had reported the inclusion of behaviour change strategies, the impact of these components was not explicitly investigated. This phase focused on defining the problem in behavioural problems; hence, the findings contributed to identifying the problem and specifying the target behaviours and population. The critical findings that guided the selection of potential target behaviours and populations in the subsequent phases were:
21.Studies often attempted to address a variety of farm risks via a single intervention.22.Multi-faceted interventions often underreported the intervention details, making it difficult to isolate the mechanism of change. 23.Very few interventions prioritised high accident-prone areas such as machines and livestock.24.Regardless of the significant role played by demographic factors such as age and farm types and the poor participation of vulnerable groups such as older farmers, interventions rarely focused on them [18,52].25.Limited interventions reported the inclusion of stakeholders’ insights in the intervention development phase.


Findings from the review highlighted the need for tailored interventions that address vulnerable populations and more narrowly targeted interventions for specific farm safety practices. Ireland has an ageing workforce, with the average age of a farmer being fifty-seven. The Irish farm safety reports that 45% of the fatal incidents on the farms involve farmers 65 years of age or older and they are reported to be eight times more vulnerable to fatal injuries than other working sectors [6,53]. While reviewing the evidence, the authors noted that various agencies and studies define “older farmers” as those over the age of 55 [54,55], 60 [56] or 65 [57], depending on the context. After considering the Teagasc’s feedback on the feasibility of recruiting older farmers for online-based interviews and future in-person activities, the lower age limit of “older farmers” was set to 60 years for the scope of the current project. The farm surveys reported that tractors and quad bikes were associated with 55% of the fatalities [41]. Hence, it was decided that the subsequent qualitative study would focus on the tractor and quad-bike-related practices among older farmers.

While a few studies employed behaviour change theories to develop interventions, most of the studies failed to report the intervention development process and components in detail. In addition, even while the intervention details were available, they were not described using the behavioural change terminologies. Hence, the authors identified the specific intervention components targeted at machine safety, analysed their descriptions, and then retrospectively coded them using the BCT taxonomy. 

### 3.2. Phase 2: Identify What Needs to Be Changed and Which Barriers and Enablers Need to Be Addressed?

Farmers identified the risky practices prevalent in the Irish farming communities along with the prevalent and potential safety practices. They discussed the types of facilitators and barriers influencing the adoption of these practices. Some of these factors were specific to certain practices, and some were related to farm safety in general. Several inductive themes related to the challenges of adopting and adhering to safety practices were identified. They were further analysed in the context of the COM-B sub-domains [33].

Several of these barriers were consistent with the findings from the literature. For example, participants repeatedly highlighted rushing and lack of situational awareness as major contributors to farm incidents. On further exploration, some common risky practices associated with them were identified, such as poor maintenance of the power take-off (PTO) shaft and its protective covers, operating the tractor without checking the perimeter and climbing off the tractor without engaging the break. However, older farmers also talked about how the factors such as their perceived self-efficacy and perceived risk associated with the tasks influenced their decision to continue or modify their farm practices as they grew older. Several participants discussed discontinuing using quad bikes since they perceived them as dangerous machinery and found them challenging to operate. 

The study also explored the farmers’ attitudes towards the potential BCTs identified in the systematic review. For example, the participants were asked about their attitudes towards mentoring programs and farm discussion groups, and they shared their recommendations for potential intervention strategies. 

### 3.3. Phase 3: Identification of Potential Target Behaviours and Intervention Components

The research team met to review the focus group findings and summarised the relevant findings from phases 1 and 2. Identifying specific barriers and facilitators and mapping them to the theoretical domain informed the identification of potentially effective intervention functions and behaviour change components. The focus group participants also suggested a few target behaviours and intervention strategies to address them, and the research team mapped them to the BCW intervention functions and BCTs using the BCW framework and BCT taxonomy V1, respectively [43].

#### 3.3.1. Phase 3.1 Co-Design Workshop

Identification of target behaviours

The relevant findings from the previous phases and the web-based survey on target behaviour categories were shared among all participants a week before the first session. Once the participants provided their consent and completed the survey by ranking the candidate behavioural change categories in terms of their relative priority, the top two were identified. Eleven participants completed the survey and selected the following two categories as the most appropriate ones: (1) allocating attention to machinery operation and the local environment and (2) installing and using appropriate safety devices on machinery.

After participants prioritised the appropriate target behaviours for the intervention via survey, the workshop started by exploring specific target behaviours under the shortlisted categories that could be potentially improved through behaviour-based interventions. While considering the specific target behaviours to be considered under the first category, participants highlighted the importance of self-evaluation of risks by farmers. They discussed increasing the habitual risk assessment in the immediate surroundings to ensure the tractor is in working order and that older people and children are not in the working area. While discussing the second category, participants discussed fitting the tractor with appropriate safety devices such as cameras and mirrors to improve visibility and awareness. They have also noted the significance of choosing appropriate and fitting implements and safety devices for farm operations; for example, the right-sized trailer and well-fitted protection covers and PPEs. After identifying the potential specific target behaviours, associated barriers and facilitators that can potentially be addressed by behaviour change-based interventions were discussed. Current participants reiterated the findings from the focus group on the prevalence of rushing and how it prompts the farmers to overlook the immediate dangers such as maintaining good conditions of equipment and a sensible pace for tractor operations. They also pointed out that raising awareness about specific risks associated with each task can encourage participants to be more alert. Likewise, the focus group participants also highlighted how the financial and time constraints along with the voluntary nature of the current safety regulations often encourage the farmers to prioritise productivity over safety. These behavioural determinants were identified as relevant for both categories.

While discussing the heterogeneous and seasonal nature of tasks, the participants raised the importance of seasonal safety messages and campaigns. Beyond the messages from the safety authorities, they described how personal stories and farm visits by friends highly resonate with their fellow farmers. 

Identification of potential intervention strategies and delivery methods.

The summarised findings and recommendations that emerged from the previous phases were shared among the participants before the online session. The first half of the discussion focused on identifying potential barriers and facilitators that can be effectively addressed through behaviour-based interventions. While the financial constraints and voluntary nature of the regulations were major concerns, the consensus was that they were more effectively addressable by financial and regulatory-based interventions. 

Participants agreed that a lack of knowledge is a major barrier and can be effectively tackled through targeted safety messages. While a few participants recommended peer-to-peer learning and buddy systems as effective strategies, others proposed marketing campaigns. One of the safety researchers explained her experience with implementing personalised safety messages through marketing campaigns. However, a few participants did raise concerns about the effectiveness of individual-level behaviour change strategies given the isolated nature of the farms and frequently reported risk habituation among farmers [6]. In the further discussion on effectively addressing these barriers, they observed that raising awareness among family members and co-workers may effectively tackle them. 

The sample presentations and surveys used for the workshops are available under the BeSafe profile in the OSF [58]. 

#### 3.3.2. Phase 3.2: Feasibility Screening with Teagasc Advisory Team

As the final step of the intervention design, multiple follow-up meetings were conducted among the authors. The intervention was designed for older farmers, but Teagasc brought up the possibility of including the younger adults in the first trial. It was agreed that, by including younger farmers, the trial could examine whether the intervention was suitable for a wider age range and whether the outcomes achieved among older farmers can be generalizable to younger farmers as well. The different perspectives and experiences of younger and older farmers regarding their health and wellbeing will be investigated since they may reveal new insights and possibilities for tailoring the intervention to meet the needs of different age groups. Hence, under the guidance of the Teagasc advisory team, it was decided that, for the first trial:
26.The length of the intervention program will be less than four hours.27.Participants from all four major farm types will be invited to the program.28.The ratio of older and younger farmers will be 50:50.


As summarised in Table A2, the target behaviours were related to improving awareness about the surrounding of a tractor, specifically blind spots of them. The first two target behaviours consisted of examining and locating the blind spots of their regular tractors, and the third behaviour involved the regular check of blind spots before starting the tractor every day. The potential target behaviours considered at various stages are summarised in Table 2. 

The summary of the findings regarding the barriers and facilitators considered at various stages is illustrated in Table 3.

Based on the findings from the first three phases, intervention functions, education, training, enablement and persuasion were finalised for the interventions. As noted in Table A3, some of the BCTs under consideration at that stage were 3.2, social reward; 13.1, identification of self as role model; 13.2, framing/reframing; 13.3, incompatible beliefs; and 16.3, vicarious consequences. However, a consensus on the operationalisation of the BCTs and delivery method was not reached by the end of the workshops. In the subsequent meetings with the Teagasc advisory team, different ways to operationalise and deliver were discussed and assessed against the feasibility criteria. For example, though the social reward was identified as a potentially effective BCT in public health safety research and endorsed by focus group participants, it was excluded based on the complexity involved in the implementation and evaluation. Another example is while considering the delivery methods for the demonstration activity, the initial consideration was one-on-one delivery of the blind spot demonstration among the farmer and his/her family members. However, given the group-based activity structure of the Irish farm programmes and the difficulty in recruiting the whole farm families, it was decided that the demonstration will be delivered in a group setting through a peer-to-peer demo instead of a farmer-to-family demo activity. 

As detailed in Table 4, the final draft of the intervention includes an in-person demo session, facilitated discussion, personalised safety training procedure and demonstration kit. The proposed active ingredients are 1.1, goal setting (behaviour); 1.2, problem solving; 1.3, goal setting (outcome); 1.4, action planning; 1.8, behavioural contract; 5.2, salience of consequences; 8.1, behavioural practice/rehearsal; and 13.1, identification of self as role model. The content and delivery method will be further refined during the dry run with a small group of stakeholders, if required.

### 3.4. Phase 4: How Can Behaviour Change Be Measured and Understood?

The detailed description of the trial design, evaluation strategies and data collection tools are available in the study protocol (pre-print) [59] and therefore, a summary is available in Table A4. In a separate publication, the findings from the feasibility study will be published, where the authors will examine how the active ingredients influenced the behaviours along with the reported feasibility, fidelity and acceptability of the program. 

## 4. Discussion

This paper describes the systematic development of an intervention to improve tractor-related safety on farms and aims to fill the lack of theoretically based and adequately described behaviour change-based interventions in the relevant literature [4,12,60]. Tractor safety is a complex process involving the interactions between tractor design, farmers’ behaviour and environmental factors [6,19,31]. The previous studies indicate that the focus of the safety interventions was on increasing the adoption of engineering solutions, introducing safety regulations and raising risk awareness [4,12]. The current study aimed to develop a safety intervention to improve the machine-related safety among Irish farmers, drawing on evidence from theoretical models, local contexts and target population. Under these criteria, a safety intervention was designed using a combination of education, persuasion, enablement and training to equip farmers with knowledge, skills, and resources to adopt preventive safety behaviours to address tractor-related blind spots at their farms and improve farm safety.

Evidence suggests that by targeting specific behaviour change mediators, the potential effectiveness of the intervention is likely to be increased [35]. Therefore, at the initial stage of the study, it was determined that it would focus on the high-accident-prone areas and vulnerable populations by looking at specific farm practices that may improve one or more areas of tractor safety. The BCW framework provided a systematic way to identify the potential intervention functions that might most likely address the enablers and facilitators, increase the adoption of target behaviours, and bring change in farmers’ safety behaviours. While the health behaviour change literature suggests various strategies to inform the selection of the intervention components, from public and patient involvement panels [27] to interviews [26], to our knowledge, there is no consensus on the most appropriate procedure. Interactions with key stakeholders and target populations provided the local context and information on specific farm practices that often lead to debilitating injuries or fatalities. Further, these interactions with the target population, key stakeholders, farm safety experts and the advisory team at various phases identified a list of the potential target behaviours and intervention components. However, the research team decided on the three specific target behaviours for the intervention based on their knowledge and recommendations of the Teagasc advisory team. For example, instead of considering the “checking the perimeter before reverse parking the tractor” as the target behaviour, “checking the blind spots before operating the tractor” was selected after these discussions. The decision was taken after considering the key criteria, such as the feasibility, safety and ethical considerations associated with the inclusion of a moving tractor in the intervention. 

The decision processes, behaviours and environmental conditions leading up to a specific farm fatal incident are complex. Hence, it is unlikely that adding one safety pre-check practice to farmers’ habits is not enough to reduce the potential dangers associated with blind spots. A combination of activities involving environment re-structuring, and raising the awareness of farmers and others farm workers and family members are required to improve the safety odds. That is why, while one of the target behaviours focused on improving the adoption of a safety pre-check into their daily practices, the other two target behaviours focused on setting up the no-visibility zone and performing the blind spot demonstration at their farms with family members or friends. 

While reviewing the existing interventions, it was noted that education and training-based interventions often target farmers alone, even though farm surveys [41] indicate that non-farmers on the farms also fall victim to fatal farm incidents. Examination of factors influencing farmers’ decisions making process repeatedly highlighted the significant role of their concerns regarding the well-being of their family members and social support [20,61,62], as family members from the family farms are often neglected [61]. These educational sessions are often conducted in a group using standard farm equipment, and takeaways are summarised in a leaflet or documents. However, the farmers and other stakeholders who participated in the current study repeatedly highlighted how unpopular paperwork and any documents, in general, are among farmers. The current study/intervention allows the participants to practice the assessment by themselves and thereafter encourages them to complete the risk assessment with the active participation of family members or co-workers on equipment that they use every day.

Before evaluating the effectiveness of a new intervention or piloting on a larger scale, it is recommended to assess the feasibility, fidelity and acceptability of the intervention [63]. A feasibility study [64] can help identify potential problems with a proposed project so that you can address them before doing a larger effectiveness trial [63,65]. The adoption of an intervention depends on the perceived acceptability because it indicates how much the intervention was thought to be appropriate by the target population [66]. If the current intervention or a part of it proves to be effective in the feasibility study, it could be introduced as part of the existing farm programmes or could proceed to a large-scale effectiveness trial.

The key findings and recommendations that will be published separately can inform future studies on the potential target behaviours, specific barriers and facilitators influencing them and potential BCTs that can be used to address and promote them effectively. 

## 5. Strengths and Limitations

The main strength of the study is the systematic development of intervention through evidence-based, theoretically informed phases and strong stakeholder engagement. This enabled transparent reporting and may enable the replication or adoption of the intervention or one of its ingredients in future studies. This is one of the few studies that addressed the older farmers’ needs while designing the intervention and ensured the participation of farmers from different age groups, thus making it suitable for both young and older farmers. Educational interventions are one of the most common strategies used by farm safety programmes; however, for the current study, the choices of the intervention components and their delivery modes were informed by the theoretical framework underpinned by behavioural models, local contexts and target population. 

Rothman [67] observed that the use of a theoretical framework provides an important conceptual and analytical framework for determining why an intervention is effective or not. The current method allowed the authors to design an evidence-based, theoretically underpinned intervention that was informed by stakeholders’ perspectives. The use of the BCW framework will also enable the study to explore the impact of each intervention component, identify the active ingredients, and conduct a detailed investigation of how the BCTs acted upon the barriers and enablers and whether it brought out the desired target behaviours or not.

The intervention focuses on a selected few target behaviours targeting the safety related to blind spots hence addressing the influencing factors associated with it. However, by describing the process in a systematic manner and reporting the findings in detail through multiple papers, the study added detailed context-specific information on prevailing machine operation and safety-related practices, factors influencing them and guidance on developing similar interventions based on the findings of the current study.

While the BCW framework gained popularity in public health research, there is limited information available on its adoption of it in the farm literature. Hence, there was a lack of available evidence to decide on the best approach to select the best target behaviours and operationalise the BCTs from the potential list of farm practices and intervention strategies identified through literature and interactions with experts and stakeholders. Therefore, the final draft of the target behaviours, selection, operationalisation and packing of intervention components were based on the researchers’ experience, feedback from the advisory team and what is likely feasible within the current research context. As observed by Cadogan et al. [26], these barriers will not be resolved until the uptake in the behaviour change-based interventions, and the detailed description of intervention development, intervention components, and their delivery become standard practice.

All the participants who contributed to the various stages of the studies attended them voluntarily. Previous studies reported that the voluntary nature of these studies often encourages only the already safety-conscious farmers to participate [68]. As a result, our findings may be subjective and may not represent the entire workforce, especially most risk-taking farmers. 

## 6. Conclusions

The study demonstrated the integration of various available evidence, such as reviews, field reports, stakeholder perspectives, and behaviour change theories, within an appropriate framework to provide a structure for integrating evidence and identifying and implementing the most appropriate behaviour change strategies. Detailed reporting of the intervention development can encourage future farm researchers to better document and report their own development process along with their key findings, thus generating a body of evidence that can be adopted as a whole or in parts by the safety agencies and researchers for developing farm safety programmes. The ongoing feasibility study will assess the potential of the intervention and modification required before progressing to a large-scale effectiveness study.

## Figures and Tables

**Table 1 ijerph-20-05394-t001:** Intervention development phases.

Phase	Tasks	Outputs	BCW Steps
PHASE 1: Describe the problem in behavioural terms	Identify the evidence-practice gap Examine the Irish farm surveys and fatality reports to gain an understanding of the local context Identify the farm groups whose behaviour needs to change Identify the areas of machine safety that need to be addressed Review the available evidence on farm machine safety Identify the BCTs present in the available safety interventions	Identified older farmers (65+) as the potential target population Identified tractors and quad bikes as the major contributors to the fatalities Identified the most commonly used BCTs and their operationalisation	Step 1—Define the problem in behavioural terms Step 5—Identify the intervention functions Step 7—Identify BCTs
PHASE 2: Identify what needs to be changed and which barriers and enablers need to be addressed?	Use qualitative methods underpinned by the COM-B model, to identify the possible target behaviours (safe farm practices) Use qualitative methods underpinned by the COM-B model to identify barriers and enablers that likely influence the target behaviours	Identified the farmers’ recommended list of target behaviours Identified the farmers’ recommended list of barriers and enablers to specific farm practices or general machine safety	Step 2—Select the target behaviour Step 4—Identify what needs to change Step 5—Identify the intervention functions Step 7—Identify BCTs
PHASE 3: Identification of potential target behaviours and intervention components 3.1 Co-design workshops 3.2 Feasibility screening with Teagasc advisory team	Identify the top two potential categories of target behaviours associated with machine-related incidents in Irish farms Identify a list of specific target behaviours under the aforementioned categories that can be influenced by behaviour change-based interventions Identify barriers and enablers that are likely to influence these specific target behaviours Identify potential behaviour change techniques and their delivery mode to overcome the barriers and enhance the enablers	Created a survey with potential categories of target behaviours Identified the top two categories Developed a list of specific target behaviours that can be targeted Developed a list of potential barriers and facilitators related to potential target behaviours Developed a list of potential ways of operationalisation and delivery method	Step 3—Specify the target behaviours Step 4—Identify what needs to change Step 5—Identify the intervention functions Step 7—Identify BCTs Step 8—Identify mode of delivery
	Identify the available evidence from the previous tasks to inform the selection of potential target behaviours, behaviour change techniques and modes of delivery Identify what is likely to be feasible, locally relevant, and acceptable and combine identified components into an acceptable intervention that can be delivered	Finalised the specific target behaviours for the intervention Finalised the BCTs, its operationalisation and delivery method into intervention components Developed the draft of the BeSafe machine safety intervention by combining the intervention components Developed and refines the materials for the intervention.	
PHASE 4: How can behaviour change be measured and understood? 4.1 Validating and refining evaluation strategy with stakeholders	Select appropriate outcome measures Determine the feasibility of outcomes to be measured	Identified the outcomes to be measured Identified the tools and methods to measure the selected outcomes	

**Table 2 ijerph-20-05394-t002:** Summary of potential target behaviours.

Specific Target Behaviour	Recommendation By *		
	Focus Group	Co-Design Workshops	Teagasc Advisory Team/Research
Whenever the farmer stops/parks the tractor, engage the handbrake securely	Y		
Farmer performs a self-risk evaluation before performing any tasks	Y	Y	
While operating a tractor, the farmer always makes sure that no one including other workers are standing near the vehicles or between thr vehicles and implements attached to it	Y		
Do not rush—farmers always perform tasks at a sensible speed taking account of working conditions and their own capabilities	Y	Y	
Always do an inspection before operating the machinery to make sure that the vehicle is in good working order	Y		
Farmers avoid phone calls while driving a tractor	Y		
Farmer always ensures that no one, including himself, stands on the farm vehicle or the implements attached to it when1. The machine is running2. The PTO rotator or any other moving parts is spinning	Y		
Farmer follows the safe hitching/unhitching procedure each time he attaches an implement to the tractor	Y	Y	
Farmer retrofits the tractor with the recommended safety devices before using it next time	Y	Y	
Farmer retrofits the roll-over protective structures specific to quad bikes before he use it next time	Y		
Always watch out especially for children and elderly persons who may cross in your path or behind you before reversing and give additional attention to blind spots	Y	Y	
Farmer always makes sure that implements are fit with the recommended safety devices before connecting them to the farm vehicle	Y		
Farmer remains alert of the immediate environment while working	Y		
Farmer makes sure to put the lap seat belt in place before taking the tractor out	Y	Y	
Farmer makes sure that a copy of the SOP is present on the vehicle where easily accessible and highly visible	Y	Y	
Prompt farmers to estimate the breaking distance of the tractors			Y
Farmers make sure to check the perimeter before reversing	Y	Y	Y

* Y indicates “Yes”.

**Table 3 ijerph-20-05394-t003:** Summary of barriers and facilitators identified.

Barriers/Facilitators	Identified as Barrier/Facilitator *	Focus Groups **	Co-Design Workshops **	Included in the Final Intervention **	COM-B Domain ***
Costly replacements/retrofitting	B	Y	Y	N	PO
Perceived poor return from investment in safety	B	Y	Y	N	PO
Changing the age old habits/reluctance to learn new ways	B	Y	Y	N	AM
Lone working environment	B	Y	Y	N	SO
Risk habituation	B	Y	Y	N	AM
Time constraints	B	Y	Y	N	PO
Safety conscious co-workers	F	Y	Y	N	SO
Low priority of upgrading the machineries	B	Y	Y	N	RM
Knowledge about the right machinery	F	Y	Y	N	PC
Stories of near misses/vicarious consequences	F	Y	Y	N	RM
Low profit margin for dry stock farmers (financial constraints)	B	Y	Y	N	PO
Farm type	B/F	Y	Y	N	PO
Safety risks are often perceived as a distant threat/risk Perception	B	Y	Y	N	RM
Priority for planning	F	Y	Y	N	RM
Lack of mandatory safety guidelines in the insurance policies	B	Y	Y	N	PO
Higher safety awareness of contractors	F	Y	Y	N	SO
Part time non-farm jobs	B	Y	Y	N	PO
Lack of accountability	B	Y	Y	N	SO
Enforcement of safety regulations (NCT certification) and mandatory built-in safety features for tractors	F	Y	Y	N	PO
Costly upgrades	B	Y	Y	N	PO
Dependency on quad bikes as a mobility device	B	Y	Y	N	PO
Seasonal use of machineries	B	Y	Y	N	PO
Ability to assess and address immediate potential risks	F	Y	N	Y	PC
Cognitive and physical health decline associated with old age	B	Y	Y	Y	PC
Recognition that the equipment/task can be dangerous	F	Y	Y	Y	RM
Knowledge of best practice	F	Y	N	Y	PC
Poor engineering standards/design	B	Y	N	Y	PO
Ineffective communication messages/materials/channels	B	Y	N	Y	PO
Guidelines—good reference for best practice	F	Y	N	Y	PO
Setting a positive role model for children/lead by example	F	Y	Y	Y	SO
Lack of discussion about safety	B	Y	N	Y	SO
Best practice—a belief that all you need is “common sense”	B	Y	N	Y	RM
Belief that accidents and risky jobs cannot be avoided in the farms	B	Y	Y	Y	RM
Willingness to learn	F	Y	N	Y	PC
Partial towards engineering and legislative solutions	B/F	Y	N	Y	RM

* B: Identified as barrier, F: Identified as facilitator, B/F: Identified as barrier and facilitator. ** Y: Yes, N: No. *** PC: Psychological capability, SO: Social opportunity, PO: Physical opportunity, RM: Reflective motivation, AM: Automatic motivation.

**Table 4 ijerph-20-05394-t004:** Intervention details.

Barriers and Enablers of Relevance Identified (Codes Identified in Focus Group)	Intervention Components	BCW Function	BCTs (* Active Ingredients)	Target Behaviours **	COM-B	Intervention Description	Expected Output	Expected Short Term Outcome
Ability to assess and address immediate potential risks (F) *** Cognitive and physical health decline associated with old age (B) *** Recognition that the equipment/task can be dangerous (F) *** Knowledge of best practice (F) Poor engineering standards/design (B) *** Ineffective communication messages/materials/channels (B) Guidelines- Good reference for best practice (F) Setting a positive role model for children/lead by example(F) Lack of discussion about safety (B) Best practice—A belief that all you need is ‘common sense’ (B) Belief that accidents and risky jobs can’t be avoided in the farms (B) Willingness to learn (F) Partial towards engineering and legislative solutions (B) ***	Estimation of the stopping distance of the tractor at various	Persuasion	5.1 Information about health consequences 5.2 Salience of consequences *	N/A	Reflective motivation: Demonstrate the consequence of standing near a moving tractor (1,2,3) Automatic motivation: Create concern about the well-being of family members (2,3)	Facilitator invite all the participants to stand near a parked tractor. Ask participants to stand where the they estimate the front of the tractor would be in 3 seconds at various speeds (5 km/hr, 20 km/hr, /50 km/hr).	Attendance Participation in the demonstration	Created negative feeling about the risk
	Demonstration of the blind spots and setting up the zone of visibility	Education Training Persuasion	4.1 Instruction on how to perform a behaviour 5.1 Information about health consequences 5.2 Salience of consequences * 6.1 Demonstration of the behaviour 8.1 Behavioral practice/rehearsal * 13.1 Identification of self as role model	1,2,3	Psychological/Physical capability: Increase the knowledge of blind spots (1,2,3) Develop skill to set up the zone of visibility (4,5) Enable to pay more attention to the immediate environment (2,3,4,5) Social Opportunity: Develop skills to a model blind spots to family members (1-6) Reflective motivation: Demonstrate the consequence of overlooking blind spots(1,2,3) Create safe parking area to protect younger family members and workers (4,5) Automatic motivation: Create concern about the well-being of family members (2,3)	Facilitator invite three participants to demonstrate blind spot of demo tractor. One participant sit on the tractor and try to locate the position of the kid sized model that the second participant is holding. Third participant mark the spots that are identified as blind spots. As participants to determine the area of no/low visibility around the tractor and set up the no visibility zone. Repeat the procedure with next 3 participants with another tractor model. The participants will be asked about to share their experience about blind spots and how blind spots differs with their family members.	Attendance Participation in the demonstration	Increased awareness of blind spots Improved skills to demonstrate the blind spots to others Improved skills to set up the zone of visibility Reminded that they can model the safety behaviour to others
	Facilitated discussion	Enablement	1.2 Problem solving * 3.2 Social support (practical) 13.1 Identification of self as role model *	1,2,3	Psychological capability: Increase knowledge on setting up the demonstration for the family members (1,2) Increase knowledge to set up the zone of visibility (3) Discussion about various strategies for the effective implementation (1,2,3) Social Opportunity: Develop skills to a model blind spots to family members(2)	Participants discuss about the demonstration experience and how they plan to complete the target behaviour.	Participation in the discussion	Reminded that they can model the safety behaviour to others Increased awareness of blind spots
	Safety training procedure	Enablement	1.1 Goal setting (behaviour) * 1.3 Goal setting (outcome) * 1.4 Action planning * 1.8 Behavioral contract * 1.9 Commitment 3.1 Social support (unspecified) 8.1 Behavioral practice/rehearsal * 8.3 Habit formation 12.1 Restructuring the physical environment 13.1 Identification of self as role model *	1,2,3	Psychological capability: Create an action plan for improving safety (1) Physical opportunity: Provision of personalised safety plan to secure the parking area (1) Social opportunity:Create social pressure to confirm with the protocol as agreed to the peers(3) Reflective motivation: Create action plans (1) Create opportunity to report their confidence (2)	Complete a tailored document for each participant based on the input from facilitated discussion. Rate their confidence on completing the activity. Participant and a peer who acts as a witness sign the contract	Individualised safety planVoluntary agreement to complete the safety goals	Demonstration of the blind spots to family members Setting up visibility zoneWalk around the tractor before starting it.
	Demonstration kit	Enablement	12.5 Adding objects to the environment	1,2	Physical opportunity: Provide the materials for the demo (1)	Provide the materials for the completion the demonstration and setting the no-visibility zone	Collection of materials	Demonstration of the blind spots to family members Setting up no visibility zone

* Identified as active ingredients; ** Target behaviours: (1) Demonstrate blind spots of tractors to family members/co-workers on their farm. (2) Mark the zone of visibility around their tractor in a parking. (3) Walk around the tractor before moving it from the parking area to ensure that nobody is near the tractor and no obstacles are present nearby; *** Participants identified these factors as directly related to the blind spots/visibility.

## Data Availability

The dataset supporting the conclusions of this article is included within the article. Additional data supporting the project is available in the OSF repository [58].

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
