# Peer review of "Increasing Machine-Related Safety on Farms: Development of an Intervention Using the Behaviour Change Wheel Approach"

_ijerph, 2023, doi:10.3390/ijerph20075394_

Round 1

Reviewer 1 Report

OVERALL

This manuscript describes the first three phases of the development of a behavioural change-based intervention with the intention of improving the safety of those who are around tractors on farms.

SPECIFIC COMMENT

Table 3   Please separate Barriers from Facilitators in the table by creating two categories within the table

_________________________________________
Addition comment

1. What is the main question addressed by the research? The study described the development of a behavioural change intervention to address tractor related blind spots at farms. According to the authors “the objective of the current study is to use the systematic Behaviour Change Wheel (BCW) intervention development approach by using existing evidence, farmer and other stakeholder perspectives to develop an evidence-based, theoretically informed intervention to increase machine-related safety on farms”. Essentially entire the study, which consists of different parts, investigates whether the BCW be used to develop and implement a feasible intervention to improve farm safety for Irish farmers by addressing tractor related blind spots on farms.

2. Do you consider the topic original or relevant in the field? Does it address a specific gap in the field? The topic is relevant as it relates to farm injuries. Tractor injuries have remained high relative over the years despite intervention and the existence of technology related solutions. Behavioural theories have been used to guide the development of interventions in medicine and agriculture. However, the systematic development of an intervention is infrequently reported. Further, for farm injuries, the use of a behavioural theory to guide the development of an intervention and evaluation of the intervention is not common.

3. What does it add to the subject area compared with other published material? The use of behavioural theory to guide the development of interventions is thought to lead to more effective interventions. A unique feature of this manuscript that the authors describe the development of an intervention and include an approach to evaluating the intervention.

4. What specific improvements should the authors consider regarding the methodology? What further controls should be considered? Line 440 and 600-603 Because the authors intend to have a 50:50 ratio of old and younger farmers in the intervention, there should also be focus groups for those under 60 to ensure that their interests are represented.

5. Are the conclusions consistent with the evidence and arguments presented and do they address the main question posed? The arguments presented are relevant to the main objective. However, a full appreciation of the decisions made with respect to the development of the BCW in the current manuscript is limited by reference to manuscripts which have not yet been published. Further the evaluation of the developed intervention will reveal the value of the approach taken, but this will be described in a separate manuscript.

6. Are the references appropriate? The references are appropriate. However, it should be noted that the that the following reference is a pre-print: Surendran, A.; McSharry, J.; Bligh, F.; McNamara, J.; Meredith, D.; O’Hora, D. Assessing the Feasibility, Fidelity and Acceptability of a Behaviour Change Intervention to Improve Tractor Safety on Farms: Protocol for the BeSafe Tractor Safety Feasibility Study. 2022

7. Please include any additional comments on the tables and figures. There are some spelling errors in the tables. For example, in Table 1 PHASE 2: “Identity” should be “Identify”. Also, Table A5, “consist” should be “consists” and “indicate” should be “indicates” For Tables 2 and 3 please indicate what is mean by “Y” and “N”

Reviewer 2 Report

Agriculture is essential work, and it is important that we look at ways to reduce serious injuries and deaths among farmers caused by machinery. This paper documents the systematic development of evidence-based, theory-based behavior change intervention to increase  machine-related safety on farm. The development of intervention measures has gone through four stages. Although it is innovative to some extent, there are still some problems to be improved:

(1) Line 77. In "2. Intervention development process", the author mainly describes the intervention development process. Is it more appropriate to put relevant content into the introduction?

(2) Line 141. ”Table 1. Intervention development phases.” “ - Identified older farmers (65+) as the target population to be targeted - Identified tractors and quad bikes as the major contributors to the fatalities - Identified the most commonly used BCTs and their operationalisation”. Why only elderly farmers are studied here? What is the basis for defining elderly farmers as those over 65 years old?

(3) Line 170-172. ”In the next phase, between January and February 2021, we conducted a qualitative study involving four semi-structured focus groups with Irish farmers above 60 years of age. Purposive sampling [42] was adopted to ensure that farmers from all four farm types (dairy, beef, sheep and tillage) were represented.” Why were the subjects over 60 years old? Were farmers under 60 years old included?

(4) Line 336-338. ”Findings from the review highlighted the need for tailored interventions that address vulnerable populations and more narrowly targeted interventions for specific farm safety practices. Ireland has an ageing workforce, with the average age of the farmer being fifty-seven.” The aging problem of farmers is described here. Is it related to the selection of farmers aged 60 or 65? Please provide a supplementary explanation.

(5) Line 440. ”3. The ratio of older and younger farmers will be 50:50.” Young farmers are mentioned here. What is the specific age structure of the object studied in this paper? Please add clarification.

(6) Line 478, and Line 482. “Table 4. Intervention details.” and “Table 4.2: Intervention details.“ The content is vague, please improve the clarity of the chart. No Table 4.1? Please check the chart number.

(7) Line 651. ”Figure A1 : Behaviour change wheel”. Please improve the size and clarity of the words in the picture.

Round 2

Reviewer 2 Report

Agree to publish in the current modified version.